# Hyperbolic Space with Hierarchical Margin Boosts Fine-Grained Learning from Coarse Labels

**Shu-Lin Xu[1], Yifan Sun[2], Faen Zhang[3], Anqi Xu[4], Xiu-Shen Wei[1]\*, Yi Yang[5]**

[1]School of Computer Science and Engineering, and Key Laboratory of New Generation Artificial Intelligence Technology and Its Interdisciplinary Applications, Southeast University
[2]Baidu Inc.    [3]AInnovation Technology Group Co., Ltd    [4]University of Toronto
[5]CCAI, College of Computer Science and Technology, Zhejiang University

## Abstract

Learning fine-grained embeddings from coarse labels is a challenging task due to limited label granularity supervision, *i.e.*, lacking the detailed distinctions required for fine-grained tasks. The task becomes even more demanding when attempting few-shot fine-grained recognition, which holds practical significance in various applications. To address these challenges, we propose a novel method that embeds visual embeddings into a hyperbolic space and enhances their discriminative ability with a hierarchical cosine margins manner. Specifically, the hyperbolic space offers distinct advantages, including the ability to capture hierarchical relationships and increased expressive power, which favors modeling fine-grained objects. Based on the hyperbolic space, we further enforce relatively large/small similarity margins between coarse/fine classes, respectively, yielding the so-called hierarchical cosine margins manner. While enforcing similarity margins in the regular Euclidean space has become popular for deep embedding learning, applying it to the hyperbolic space is non-trivial and validating the benefit for coarse-to-fine generalization is valuable. Extensive experiments conducted on five benchmark datasets showcase the effectiveness of our proposed method, yielding state-of-the-art results surpassing competing methods.

## 1 Introduction

In deep learning, rich supervised information is crucial for the generalization ability of deep models. Fine-grained visual recognition [36] is a fundamental problem in computer vision, which requires accurately identifying subordinate (fine-grained) categories within the same meta (coarse-grained) category. However, in many domains, annotating fine-grained data requires the involvement of domain experts, such as in pipeline failure detection [1], biodiversity recognition [30] and medicine analyses [25], leading to high annotation costs or even infeasibility. In contrast, obtaining the coarse labels, *e.g.*, the meta category of a species in biodiversity recognition, is much easier. In such cases, we attempt to use coarse-grained labels to train models and apply them to finer-grained recognition tasks. Such a task is challenging as it lacks the detailed supervisory information required for fine-grained tasks, preventing us from obtaining sufficiently detailed label information to train models to recognize fine-grained features and distinctions. In this paper, we attempt to train models using coarse-grained labels and apply them to challenging few-shot fine-grained recognition tasks, which are highly demanding and have practical significance.

---

\*Corresponding author. This work was supported by National Key R&D Program of China (2021YFA1001100), National Natural Science Foundation of China under Grant (62272231), Natural Science Foundation of Jiangsu Province of China under Grant (BK20210340), the Fundamental Research Funds for the Central Universities (NJ2022028), and CAAI-Huawei MindSpore Open Fund.

To tackle these challenges, we propose a novel method termed Poincaré embedding with hierarchical cosine margins (PE-HCM). Specifically, PE-HCM embeds samples into a Poincaré embedding (which is also known as an embedding in the hyperbolic space [21]) and enhances their discriminative ability with a hierarchical cosine margin manner. Our PE-HCM has two coupled features, *i.e.*, hyperbolic space and hierarchical cosine margins:

- *Hyperbolic space.* we first embed the samples into the hyperbolic space instead of the regular Euclidean space, as the hyperbolic space provides stronger expressive power for hierarchical relationships [19]. Since the tasks require an embedding space that can not only capture the coarse-grained information provided during training but also generalize well to fine-grained categories during the test, the embedding space should cover multiple granularities from coarse to fine. The hyperbolic space well satisfies this prerequisite with its natural hierarchical structure.

- *Hierarchical consine margins.* Based on the hyperbolic space, we further incorporate hierarchical cosine distance constraints to impose a hierarchical proximity relationship among sample pairs. We construct finer-grained labels than the coarse-grained training labels through unsupervised approaches such as data augmentation and clustering. Specifically, similar to SimCLR [4], we apply two sets of data augmentations to a batch of data. For the same sample, different augmentations are assigned with the same instance-level label, while different samples are assigned with different instance-level labels. By clustering the samples within a coarse-grained category, we obtain multiple clusters that represent the fine-grained categories between the coarse-grained and the instance-level categories. As a result, we have instance-level, fine-grained, and coarse-grained labels. Correspondingly, there are four types of pairwise relationships: same instance-level category, same fine-grained category, same coarse-grained category, and different coarse-grained categories.

Moreover, to derive the appropriate target distances that will be used for different levels, we propose an adaptive strategy to update the target distances during training. It utilizes the average distance between sample pairs in each batch to update the target cosine distances with momentum. The adaptive strategy plays a crucial role in ensuring that the target distances reflect the real data distribution. By dynamically adjusting the target distances during training, our method can better capture the underlying relationships and adapt to the fine-grained characteristics of the data. In a batch of training data, we derive the target cosine distance distributions for the pairwise relationships based on their category relationships in the hyperbolic embedding space. By enforcing the consistency between the feature distribution and the category relationship branch, we enhance the hyperbolic embedding space's generalization and discriminative capability for implicit finer-grained categories.

In experiments, we perform PE-HCM on five popular benchmark datasets, *i.e.*, CIFAR-100 [16] and four sub-datasets {LIVING-17, NONLIVING-26, ENTITY-13, ENTITY-30} from BREEDS [23]. Our method has achieved state-of-the-art recognition accuracy on these datasets, thereby demonstrating its effectiveness and its potential for practical applications.

In summary, our major contributions are three-fold:

- We propose a novel method that addresses the challenging task of fine-grained learning from coarse-grained labels. It bridges the gap between coarse-grained and fine-grained labels and leverages the knowledge learned from coarse-grained categories for fine-grained recognition tasks.

- We develop a hyperbolic embedding with hierarchical cosine margins (HCM). HCM enforces relatively large/small similarity margins between coarse/fine classes. The key point of HCM, *i.e.*, the target similarity margins, are dynamically updated using an adaptive strategy.

- We conduct comprehensive experiments on five popular benchmark datasets, and our proposed method achieves superior recognition accuracy over competing solutions on these datasets.

## 2   Related Work

**Coarse and Fine Learning**   Coarse and fine learning has been an important topic in computer vision and machine learning. Numerous methods [2, 7, 8, 26, 27, 29, 35, 40, 42] and theoretical studies [10] have been proposed to address this problem, with the goal of leveraging coarse-grained labeled data to improve fine-grained recognition. On the one hand, several methods have been proposed to tackle the coarse and fine learning problem. For instance, Stretcu *et al.* [26] proposed a coarse-to-fine curriculum learning method that can dynamically generate training sample sequences

based on task difficulty and data distribution, thereby accelerating model convergence and improving generalization ability. Xiang *et al.* [40] proposed a coarse-to-fine incremental few-shot learning method that can use coarse-grained labels to perform contrastive learning on the embedding space and then used fine-grained labels to normalize and freeze the classifier weights, thereby solving the class incremental problem. Sun *et al.* [27] developed a dynamic metric learning method that can adaptively adjust the metric space according to different semantic scales, thereby improving the performance of multi-label classification and retrieval. Cui *et al.* [7] designed a coarse-to-fine pseudo-labeling guided meta-learning method that can use coarse labels to generate pseudo-labels, and updated model parameters through meta-optimizer, thereby achieving fast adaptation in few-shot classification tasks. Bukchin [2] proposed a fine-grained angular contrastive learning method with coarse labels that can use coarse labels as prior knowledge to constrain the angular loss function, and constructed positive and negative sample pairs through random sampling and data augmentation, thereby achieving robust feature representation in fine-grained image recognition tasks. On the other hand, Fotakis *et al.* [10] provided a theoretical analysis of the generalization error of learning with coarse labels, which can recover the true distribution statistically without requiring additional information or assumption conditions.

Overall, previous work on coarse and fine learning demonstrates the significance of this problem and highlights the potential benefits of leveraging coarse-grained labeled data for fine-grained recognition.

**Few-Shot Learning**   Few-shot learning [9, 20, 24, 28, 31, 33, 37–39, 41, 44–46] has become a popular research direction in computer vision due to the difficulty of collecting and annotating large datasets. The goal of few-shot learning is to enable models to quickly acquire knowledge from a small number of new samples, which is particularly suitable for fine-grained recognition as fine-grained level annotations are often costly. To address this problem, various methods have been proposed, which can be broadly categorized into three types: metric-based methods, optimization-based methods, and attention-based methods. In concretely, metric-based methods, *e.g.*, prototypical networks [24], relation networks [28], matching networks [31], and feature generating networks [39], learn a metric space where the similarity between support and query samples is computed. These methods aim to learn a feature representation that captures the discriminative information of categories and a distance metric that can compare samples in this feature space. Optimization-based methods, *e.g.*, model-agnostic meta-learning [9] and piecewise classifier mappings [37], aim to learn a model that can quickly adapt to new categories with only a few examples. These methods learn an initialization that can be fine-tuned quickly on a new task with a small amount of data. Attention-based methods, *e.g.*, MultiAtt [33], MattML [46] and Dual Att-Net [41], use attention mechanisms to identify the most informative parts or regions of the input images. These methods aim to learn a feature representation that is not only discriminative but also informative for few-shot recognition.

In summary, few-shot learning has been extensively studied in recent years, and various methods have been proposed to tackle this problem. While the aforementioned papers have shown success in addressing the problem, they still rely on a considerable amount of labels with the same granularity as the testing granularity during training. In contrast, our work aims to learn fine-grained recognition only from coarse labels during training, which is a more challenging task.

**Hyperbolic Geometry**   In the field of deep learning, the hyperbolic space was first proposed by Nickel and Kiela [19] to learn hierarchical representations of symbolic data such as text and graphs by embedding them into an $n$-dimensional Poincaré ball, which showed that hyperbolic embeddings can capture the hierarchical relationships in knowledge graphs more effectively than Euclidean embeddings. Similarly, Ganea *et al.* [11] proposed a hyperbolic neural network for modeling tree-structured data, which was observed to outperform Euclidean-based models. In addition, Sala *et al.* [22] utilized the hyperbolic space to embed words in natural language processing tasks. Khrulkov *et al.* [15] proposed a hyperbolic image embedding method to represent hierarchical structures in images.

Compared to the Euclidean space which has a constant sectional curvature of $0$, the hyperbolic space has a constant negative curvature, which enables more efficient low-dimensional embedding for modeling hierarchical structure property data [19]. As illustrated in the left of Fig. 1, in this Poincaré disk, the exponential growth of distances matches the exponential growth of nodes with the tree depth [11]. This property has been shown to be particularly powerful for representing tree-structured data, where the hierarchy of the data can be more naturally captured in the hyperbolic space.

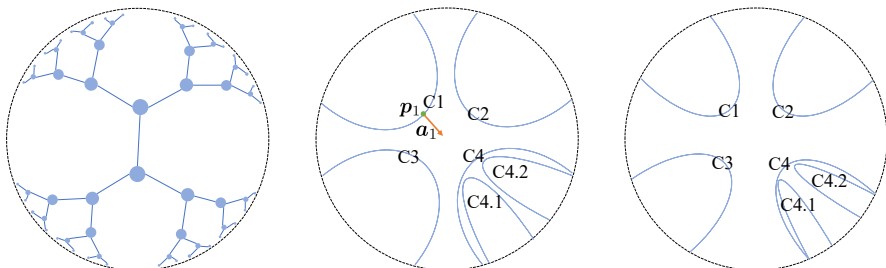

Figure 1: **Left**: An example of modeling a regular tree using a 2D Poincaré disk, where all connected nodes are equally spaced apart. **Middle**: Decision boundary in a Poincaré disk, where $p_1$, $a_1$ are a point and a normal vector determining the decision boundary, and the region bounded by the arc and the disk represents a category. C1, C2, C3 and C4 represent four coarse-grained categories, while C4.1 and C4.2 represent fine-grained subcategories of category C4. **Right**: Decision boundary with the cosine distance constraint in a Poincaré disk.

In contrast, Euclidean space has a flat geometry, which limits its ability to represent hierarchical structures efficiently.

## 3  Methodology

**Task Formulation**  In the coarse-to-fine setting [2], our goal is to train the model using only coarse-grained labels during training and achieve strong generalization performance on fine-grained categories. To evaluate the model's coarse-to-fine capability, we test it on the challenging few-shot fine-grained recognition tasks, where the model needs to recognize fine-grained categories with limited training samples. In concretely, let $\mathcal{Y}_{coarse} = \{y_1, y_2, \ldots, y_{N_{coarse}}\}$ be a set of coarse-grained classes (*e.g.*, `cat`, `dog`, `bird`), and $\mathcal{Y}_{fine} = \{y_1', y_2', \ldots, y_{N_{fine}}'\}$ be a set of fine-grained classes (*e.g.*, `husky`, `samoyed`, `burmese`). Given an auxiliary training set $\mathcal{B}$, it contains $N_{\mathcal{B}}$ labelled training images $\mathcal{B} = \{(I_1, y_1), (I_2, y_2), \ldots, (I_{N_{\mathcal{B}}}, y_{N_{\mathcal{B}}})\}$, where $I_i$ is an example image and $y_i \in \mathcal{Y}_{coarse}$ is its corresponding label. Our goal is to use $\mathcal{B}$ to learn a model that recognizes different fine-grained patterns when given a small number (*e.g.*, 1 for each class, as assumed in this paper) of novel fine-grained labeled samples $\mathcal{N} = \{(I_1', y_1'), (I_2', y_2'), \ldots, (I_{N_{\mathcal{N}}}', y_{N_{\mathcal{N}}}')\}$, where $I_i'$ is a support image and $y_i' \in \mathcal{Y}_{fine}$ is its corresponding label. If $\mathcal{N}$ contains $N$ categories, each with $K$ support images, then it is usually regarded as an $N$-way $K$-shot task.

**Recap of the Hyperbolic Space**  We use the Poincaré ball [19] to perform the hyperbolic space. Formally, an $n$-dimensional Poincaré ball is defined by a manifold $\mathbb{B}^n = \{z \in \mathbb{R}^n : c\|z\| < 1, c > 0\}$, where $\|\cdot\|$ denotes the Euclidean norm and the hyperparameter $c$ is used to modify the curvature. $\mathbb{B}^n$ is an open ball of radius $1/\sqrt{c}$, and with $c \to 0$, $\mathbb{B}^n$ is close to the regular Euclidean space.

In the application of the hyperbolic space, hyperbolic neural networks [11] were first proposed to adopt the formalism of Möbius gyrovector spaces to define the hyperbolic versions of feed-forward networks and Multinomial Logistic Regression (MLR) [11]. In concretely, after using the traditional neural network (*e.g.*, CNN) with Multi-Layer Perceptron (MLP) to get an Euclidean representation vector $x \in \mathbb{R}^n$, we map it to hyperbolic representation using the *exponential* map:

$$z = \exp^c(x) = x \tanh(\sqrt{c}\|x\|)/(\sqrt{c}\|x\|), \tag{1}$$

where $\exp^c(\cdot)$ donates the exponential mapping with curvature $c$ and $\tanh(\cdot)$ refers to the hyperbolic tangent function. Then, we use a hyperbolic hyperplane

$$H_{p_k, a_k} = \{x \in \mathbb{R}^n : \langle -p_k + x, a_k \rangle = 0\}, \tag{2}$$

determined by two learnable vectors $p_k, a_k \in \mathbb{R}^n$ as the decision boundary (*e.g.*, $p_1$ and $a_1$ as shown in the middel of Fig. 1) to predict the category $k$:

$$p(y = k | z) \propto \exp(\text{sign}(\langle -p_k + z, a_k \rangle)\|a_k\|d(z, H_{p_k, a_k})), \tag{3}$$

where the $\langle \cdot, \cdot \rangle$ represents the inner product, $a_k$ is a normal vector to $H_{p_k, a_k}$ and $d(z, H_{p_k, a_k})$ is the hyperbolic space distance [11] from point $z$ to the hyperplane $H_{p_k, a_k}$.

**Motivation**  We hereby present the motivation behind our work. In recent years, there have been a lot of efforts to explore the hyperbolic space to make it better suitable for a variety of different

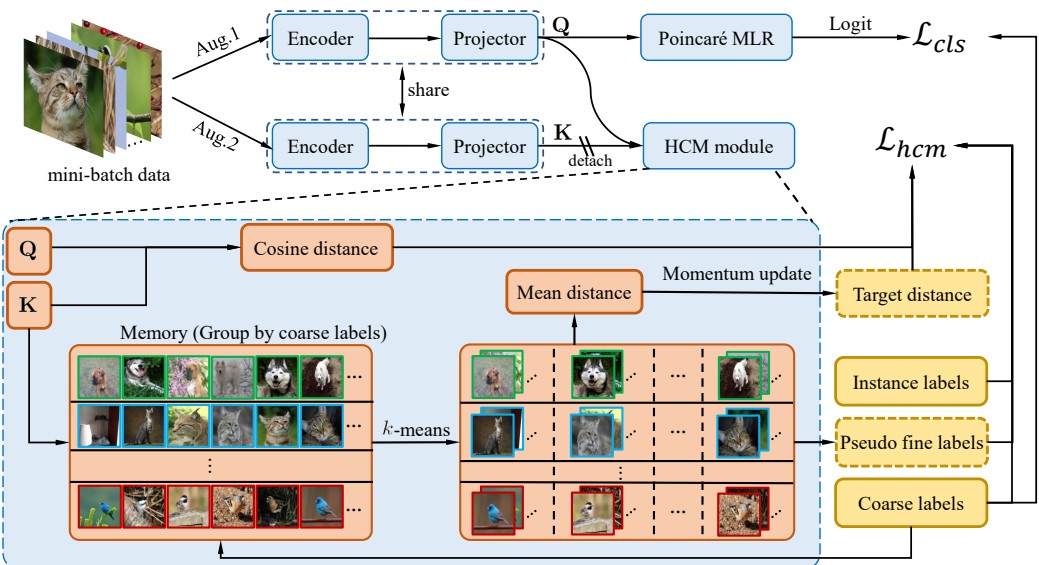

Figure 2: Illustration of our method during training. Our method uses two views of the same input images to obtain the representations $\mathbf{Q} = \{\boldsymbol{q}_1, \boldsymbol{q}_2, \ldots, \boldsymbol{q}_{N_B}\}$ and $\mathbf{K} = \{\boldsymbol{k}_1, \boldsymbol{k}_2, \ldots, \boldsymbol{k}_{N_B}\}$ via an encoder and a projector. The representations $\mathbf{Q}$ are classified into coarse classes by the Poincaré Multinomial Logistic Regression (MLR) head, which is followed by the cross-entropy loss $\mathcal{L}_{cls}$. We calculate the cosine distance between each pair of examples (*e.g.*, $\boldsymbol{q}_i$ and $\boldsymbol{k}_j$), and use the corresponding target distance to constrain it via $\mathcal{L}_{hcm}$ according to their class hierarchical relationship. We calculate the average hierarchical distance in each mini-batch to carry out momentum updates to the target distance. By using coarse-grained labels for supervised learning in a hyperbolic space and imposing hierarchical cosine distance constraints on features, we can obtain more discriminative representations for unlabeled fine-grained classes.

applications [6, 11, 14, 15, 19, 22, 34, 43]. Some efforts strive [6, 34] to obtain large-margin decision boundaries to enhance the expressive power of hyperbolic models. However, since these methods directly perform complex operations in the hyperbolic space and cannot be extended to multi-grained label tasks (*e.g.*, an object having `Labrador`, `dog` and `mammal` labels at the same time), these methods cannot make good use of the potential of the hyperbolic space to deal with multi-grained label data.

In the regular Euclidean space, researchers have proposed many methods [17, 32] based on the cosine distance to increase the angular boundary between different categories, which brings a considerable improvement to the performance of the model. Therefore, we introduce the cosine distance constraint into the hyperbolic space, and as far as we know, we are the first to do so in the hyperbolic space. According to Eq. (1), it is known that a vector does not change its direction after being mapped from the Euclidean space to the hyperbolic space by *exponential* mapping. This property, aka Conformality [21], plays a crucial role in our context. It ensures that the angles between curves or vectors in the Euclidean space remain consistent when projected onto the Poincaré disk. Therefore, as the comparison between the middle and right images in Fig. 1, the introduction of large-margin cosine distance constraint in the hyperbolic space should have similar properties as introducing the cosine distance constraints in the Euclidean space. After that, it could make the decision boundaries have larger / small similarity margins between coarse/fine classes, which enhances the generalization and discriminative capabilities of the hyperbolic space for fine-grained categories.

**Overall Framework** The method we proposed aims at learning a mapping function $f_{embed}(\cdot)$ using only coarse-grained label training data that embeds raw inputs into a hyperbolic space with a hierarchical distance structure for fine-grained visual recognition. Fig. 2 depicts our method at the training time, where our method is termed as Poincaré embedding with hierarchical cosine margins (PE-HCM). Specifically, we first introduce two separate random augmentations for mini-batch examples to get two correlated views, and then obtain two groups of representations $\mathbf{Q}$ and $\mathbf{K}$ by an encoder and a projector. For $\mathbf{Q}$ and $\mathbf{K}$ these two groups sample points in embedding spaces, we respectively use Poincaré Multinomial Logistic Regression (MLR) and our proposed Hierarchical Cosine Margin (HCM) manner to optimize the embedding space.

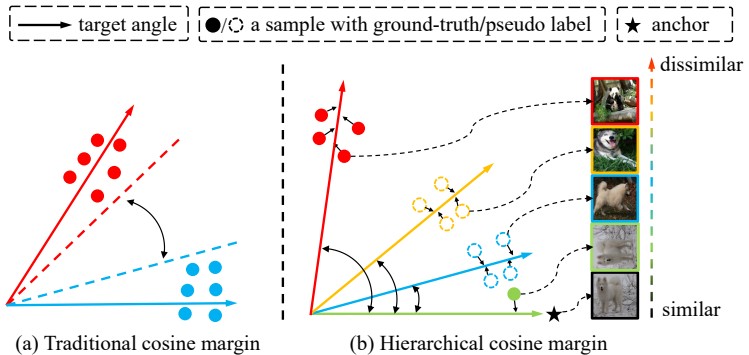

Figure 3: Comparisons between traditional and hierarchical cosine margins. Different colors indicate different categories. (a) In the latent space, we just need to make a decision boundary with a large-margin cosine distance for each pair of classes. (b) Sample pairs with three-grained labels correspond to four-level complex hierarchical target cosine distances. For instance, a pair of examples can belong to the same instance, the same fine-grained class, the same coarse-grained class, or two different coarse-grained classes.

Concretely, the Poincaré MLR module first predicts the softmax probability distribution of coarse-grained categories. Then, we use the coarse-grained labels $y$ for supervised learning

$$\mathcal{L}_{cls} = \text{CrossEntropy}(p(y = k|\boldsymbol{x}), y), \tag{4}$$

where $\text{CrossEntropy}(\cdot)$ is the cross-entropy loss and $p(y = k|\boldsymbol{x})$ is the probability prediction distribution as Eq. (3). Based on this, $f_{embed}(\cdot)$ can separate different coarse-grained samples in the embedding space.

**Hierarchical Cosine Margin Manner**    On this basis, the HCM manner constrains the distance distribution between sample pairs to be consistent with the hierarchical distribution of sample labels, so that sample pairs with different grained labels have different gaps in the embedding space, thus obtaining a more refined hierarchical embedding space that is conducive to fine-grained recognition. Specifically, we first group features $\mathbf{K}$ according to coarse-grained labels and store samples with the same coarse-grained labels in the same group in memory. We keep the latest $M$ samples for each coarse-grained category in memory. Then, we perform $k$-means clustering on each group of features respectively, dividing each group of sample features into smaller groups of clusters. We regard these more subdivided clusters as fine-grained categories under coarse-grained categories and give different clusters different fine-grained pseudo-labels. From this point on, a feature in the embedding space has three levels of granularity labels: instance-level label (each input sample has its own instance-level label, a sample's different augmentations have the same instance-level label), fine-grained pseudo-labels and coarse-grained labels. Therefore, for a feature vector of an anchor sample in $\mathbf{Q}$, there are four hierarchical relationships in $\mathbf{K}$: same instance-level label, same fine-grained label, same coarse-grained label, and different coarse-grained labels.

The hierarchical cosine distance distribution is demonstrated in Fig. 3(b). From bottom to top, the visual similarity between the samples and the anchor decreases, resulting in larger cosine distances between the samples and the anchor in the embedding space.

This creates a hierarchical cosine distance distribution that follows the same pattern as the visual similarity distribution between categories. Our proposed HCM manner is more refined and complex than the traditional cosine distance constraint, as shown in Fig. 3(a). The traditional cosine distance only needs to consider two types of relationships, *i.e.*, similar or dissimilar, while the HCM manner needs to consider the four hierarchical similarity relationships mentioned above in our setting.

In order to constrain the sample features by the hierarchical cosine distance, we calculate the cosine distance distribution between each feature $\{\boldsymbol{q}_i\}_{i=1}^{N_B}$ in $\mathbf{Q}$ and all features $\{\boldsymbol{k}_j\}_{j=1}^{N_B}$ in $\mathbf{K}$ by

$$\boldsymbol{W}_i = [d_{cos}(\boldsymbol{q}_i, \boldsymbol{k}_1), d_{cos}(\boldsymbol{q}_i, \boldsymbol{k}_2), \ldots, d_{cos}(\boldsymbol{q}_i, \boldsymbol{k}_{N_B})], \tag{5}$$

where $d_{cos}(\boldsymbol{u}, \boldsymbol{v}) = 1 - cos(\theta) = 1 - \frac{\boldsymbol{uv}}{\|\boldsymbol{u}\|\|\boldsymbol{v}\|} \in [0, 2]$ is a cosine distance function, and $N_B$ is the training batch size. As shown in Fig. 3(b), the distance between two samples in the embedded space becomes larger as the category gap between them increases. For example, when the relationship between sample pairs is the same physical object, different objects of the same kind, different species of the same genus, and different genera of the same family, the sample pairs become less and less

similar. To simplify the model, we set the distances between sample pairs with the same sample, the same fine-grained category, the same coarse-grained category, and different coarse-grained categories as $d_0, d_1, d_2,$ and $d_3$, respectively, where $d_0 < d_1 < d_2 < d_3$. We obtain the target semantic distance between each feature $\{\boldsymbol{q}_i\}_{i=1}^{N_B}$ in $\mathbf{Q}$ and all features $\{\boldsymbol{k}_j\}_{j=1}^{N_B}$ in $\mathbf{K}$ for each sample pairs by

$$\boldsymbol{M}_i = [d_t(\boldsymbol{q}_i, \boldsymbol{k}_1), d_t(\boldsymbol{q}_i, \boldsymbol{k}_2), \dots, d_t(\boldsymbol{q}_i, \boldsymbol{k}_{N_B})], \tag{6}$$

where $d_t(\boldsymbol{u}, \boldsymbol{v}) = \begin{cases} d_0 & \text{if } \boldsymbol{u}, \boldsymbol{v} \text{ are same instance-level labels,} \\ d_1 & \text{else if } \boldsymbol{u}, \boldsymbol{v} \text{ are same fine-grained pseudo labels ,} \\ d_2 & \text{else if } \boldsymbol{u}, \boldsymbol{v} \text{ are same coarse-grained labels,} \\ d_3 & \text{otherwise.} \end{cases}$ . To ensure that the

feature distance distribution of samples in the embedding space aligns with the hierarchical class semantic similarity, we approximate the distributions of the pairwise feature cosine distances of samples and targets using the Kullback-Leibler (KL) divergence

$$\mathcal{L}_{hcm} = \frac{1}{N_B} \sum_{i=1}^{N_B} \text{KL}(\boldsymbol{M}_i, \boldsymbol{W}_i). \tag{7}$$

Finally, we combine Eq. (4) and Eq. (7) to obtain the overall loss during the training process

$$\mathcal{L} = \mathcal{L}_{cls} + \alpha \mathcal{L}_{hcm}, \tag{8}$$

where $\alpha$ is a trade-off hyperparameter.

During testing, we utilize the trained $f_{embed}(\cdot)$ to embed the support set and query set samples into the hyperbolic space. In the following, we classify the query samples by computing their distance to the support samples through the $k$-nearest neighbors method.

**Adaptive Hierarchical Cosine Distance** In Eq. (6), we use $[d_0, d_1, d_2, d_3]$ in $d_t(\cdot)$ as the target values of the feature cosine distance distribution, which should be determined by the intrinsic hierarchy in different datasets, so we propose an Adaptive Hierarchical Cosine Distance (AHCD) strategy to update the target values during the training process according to the training data. Specifically, we set the target value $d_0$ of samples with the same instance-level label to 0, and the target value $d_3$ of samples with different coarse-grained labels to 1 which correspond to angles of $90°$. We initialize $d_1$ and $d_2$ to 0.134 and 0.5 which correspond to angles of $30°$ and $60°$, respectively, and using $d_0$ and $d_3$ as anchor values to uniformly distribute them. During training, we calculate the average feature cosine distance between different examples with the same grained label in each batch of data

$$\bar{d}_l = \frac{1}{N_l} \sum_{\boldsymbol{q}_i, \boldsymbol{k}_j \in \Omega_l}^{N_l} d_{cos}(\boldsymbol{q}_i, \boldsymbol{k}_j), \tag{9}$$

where $l \in \{1, 2\}$ respectively represent fine-grained label hierarchy and coarse-grained hierarchy (*i.e.*, $\bar{d}_1$ and $\bar{d}_2$ respectively represent average distance of sample pairs with same fine-grained label and same coarse-grained label), $\Omega_l$ represents a set of the sample pairs that belong to same grained labels, $N_l$ is the size of $\Omega_l$. Then, we update the target semantic distance by momentum update

$$d_l = \beta d_l + (1 - \beta)\bar{d}_l. \tag{10}$$

We use the momentum update method to dynamically update the target semantic cosine distance of sample pairs with the same fine-grained label and the same coarse-grained label, so that $d_1$ and $d_2$ can adjust values adaptively according to the distribution of data in different datasets.

## 4 Experiments

**Benchmark Datasets** Table 1 summarizes the benchmark datasets CIFAR-100 [16] and BREEDS [23]. The BREEDS dataset includes four subsets derived from ImageNet, namely LIVING-17, NONLIVING-26, ENTITY-13, and ENTITY-30, each of which has a class hierarchy calibrated to ensure that classes at the same level have similar visual granularity.

Table 1: Summaries of the benchmark datasets. L-17, NL-26, E-13, E30 are the LIVING-17, NONLIVING-26, ENTITY-13, ENTITY-30 sub-datasets from BREEDS [23].

| Datasets | L-17 | NL-26 | E-13 | E-30 | CIFAR-100 |
|---|---|---|---|---|---|
| # Coarse classes | 17 | 26 | 13 | 30 | 20 |
| # Fine classes | 68 | 104 | 260 | 240 | 100 |
| # Train images | 88K | 132K | 334K | 307K | 50K |
| # Test images | 3.4K | 5.2K | 13K | 12K | 10K |
| Image resolution | 224 | 224 | 224 | 224 | 32 |

Table 2: Comparisons on BREEDS [23]. Red and blue bold numbers are the top two best results.

| Methods | LIVING-17 | | NONLIVING-26 | | ENTITY-13 | | ENTITY-30 | |
|---|---|---|---|---|---|---|---|---|
| | 5-way | all-way | 5-way | all-way | 5-way | all-way | 5-way | all-way |
| Fine upper-bound | $90.75{\pm}0.48$ | $62.65{\pm}0.18$ | $90.33{\pm}0.47$ | $60.68{\pm}0.14$ | $94.72{\pm}0.33$ | $65.18{\pm}0.09$ | $94.02{\pm}0.36$ | $63.72{\pm}0.10$ |
| MoCo-v2 | $56.66{\pm}0.70$ | $18.57{\pm}0.11$ | $63.51{\pm}0.75$ | $21.07{\pm}0.11$ | $82.00{\pm}0.67$ | $33.06{\pm}0.07$ | $80.37{\pm}0.62$ | $28.62{\pm}0.06$ |
| MoCo-v2-ImageNet | $82.21{\pm}0.73$ | $40.29{\pm}0.14$ | $77.07{\pm}0.78$ | $34.78{\pm}0.13$ | $85.24{\pm}0.60$ | $35.62{\pm}0.08$ | $83.06{\pm}0.62$ | $31.73{\pm}0.08$ |
| SWAV-ImageNet | $79.83{\pm}0.65$ | $38.79{\pm}0.15$ | $76.26{\pm}0.71$ | $33.94{\pm}0.11$ | $81.15{\pm}0.65$ | $33.57{\pm}0.07$ | $79.91{\pm}0.54$ | $31.15{\pm}0.07$ |
| ANCOR | $89.23{\pm}0.55$ | $45.14{\pm}0.12$ | $86.23{\pm}0.54$ | $43.10{\pm}0.11$ | $90.58{\pm}0.54$ | $42.29{\pm}0.08$ | $88.12{\pm}0.54$ | $41.79{\pm}0.08$ |
| ANCOR-$fc$ | $90.41{\pm}0.57$ | $46.19{\pm}0.16$ | $88.77{\pm}0.54$ | $45.34{\pm}0.13$ | $89.05{\pm}0.58$ | $38.52{\pm}0.08$ | $91.84{\pm}0.49$ | $42.33{\pm}0.10$ |
| SCGM-G | $89.72{\pm}0.54$ | $48.74{\pm}0.15$ | $89.87{\pm}0.51$ | $49.25{\pm}0.13$ | $90.15{\pm}0.51$ | $40.00{\pm}0.08$ | $92.90{\pm}0.46$ | $42.17{\pm}0.08$ |
| SCGM-A | $90.97{\pm}0.55$ | $49.31{\pm}0.16$ | $88.78{\pm}0.55$ | $46.93{\pm}0.13$ | $88.48{\pm}0.59$ | $41.07{\pm}0.09$ | $91.22{\pm}0.51$ | $44.14{\pm}0.09$ |
| **Ours** | $90.94{\pm}0.43$ | $53.09{\pm}0.11$ | $89.97{\pm}0.42$ | $50.12{\pm}0.11$ | $91.24{\pm}0.38$ | $41.64{\pm}0.09$ | $92.95{\pm}0.40$ | $44.53{\pm}0.09$ |

Table 3: Comparisons on CIFAR-100 [16]. Red and blue bold numbers are the top two best results.

| Methods | 5-way | all-way |
|---|---|---|
| Fine upper-bound | $75.53{\pm}0.68$ | $31.35{\pm}0.11$ |
| ANCOR | $74.56{\pm}0.70$ | $29.84{\pm}0.11$ |
| ANCOR-$fc$ | $74.73{\pm}0.73$ | $27.32{\pm}0.10$ |
| SCGM-G | $76.19{\pm}0.73$ | $29.92{\pm}0.11$ |
| SCGM-A | $77.37{\pm}0.77$ | $25.91{\pm}0.10$ |
| **Ours** | $81.42{\pm}0.69$ | $36.28{\pm}0.12$ |

Table 4: Comparisons of intra-class (all fine-grained class recognition within a random coarse class) on the BREEDS [23]. Red and blue bold numbers are the top two best results.

| Methods | Living17 | Nonliving26 | Entity13 | Entity30 |
|---|---|---|---|---|
| Fine upper-bound | $70.72{\pm}0.92$ | $74.02{\pm}0.91$ | $72.24{\pm}0.60$ | $73.86{\pm}0.67$ |
| ANCOR | $48.77{\pm}0.71$ | $49.64{\pm}0.88$ | $42.00{\pm}0.47$ | $45.17{\pm}0.59$ |
| ANCOR-fc | $51.07{\pm}0.82$ | $53.51{\pm}1.00$ | $41.83{\pm}0.48$ | $47.82{\pm}0.65$ |
| SCGM-G | $53.48{\pm}0.81$ | $57.32{\pm}1.04$ | $43.89{\pm}0.58$ | $46.80{\pm}0.78$ |
| SCGM-A | $53.88{\pm}0.90$ | $55.12{\pm}1.00$ | $45.09{\pm}0.58$ | $50.02{\pm}0.71$ |
| **Ours** | $57.11{\pm}0.93$ | $58.63{\pm}1.01$ | $44.85{\pm}0.54$ | $51.24{\pm}0.65$ |

**Baselines**  We compare PE-HCM with the most relevant state-of-the-art models on embedding learning: (1) MoCo-v2 [5], trained on the above datasets; (2) MoCo-v2-ImageNet [5], pre-trained on the official full ImageNet; (3) SWAV-ImageNet, pre-trained by Caron [3]; (4) ANCOR [2], combine supervised and self-supervised contrastive learning with their proposed Angular normalization module; (5) SCGM [18], based on the superclass-conditional Gaussian mixture model. (6) 'Fine upper-bound' [2], natualy trained on the fine-grained labels.

**Implementations**  For fair comparisons with ANCOR [2] and SCGM [18], we use ResNet-12 [12] and ResNet-50 [12] as the backbone network for CIFAR-100 [16] and BREEDS [23]. We use a three-layer MLP as the projector, and the exponential mapping follows it to embed the features in the hyperbolic space. The input, hidden, and output layer dimensions of the MLP are $a \rightarrow a \rightarrow b$, where $a$ is 640 and 2048 for ResNet-12 and ResNet-50, respectively, and the output layer dimension $b$ is 128, following ANCOR and SCGM. For the hyperparameters, we set $c = 0.001$ in Eq. (1), $\alpha = 800$ in Eq. (8), $\beta = 0.999$ in Eq. (10). We used the Adam optimizer to train the model on 4 GeForce RTX 3090 Ti GPUs, and training a total of 200 epochs. For CIFAR-100 and BREEDS, the batch size were 1024 and 256, the initial learning rates were $5 \times 10^{-4}$ and $1.25 \times 10^{-4}$, the learning rates were reduced by 10 times when the epoch was 120 epoch and 160 epoch. In order to prevent the distance distribution from getting stuck in local optima, we reinitialize $d_1 = 0.134$ and $d_2 = 0.5$ at the 120-th epoch and 160-th epoch. We followed ANCOR [2] to implement random data enhancement using random resized crop, random horizontal flipping, random color Jitter, random grayscale, and random Gaussian smoothing during training. We evaluate the performance using 5-way and all-way 1-shot settings during testing. The evaluation is conducted on 1000 random episodes, and we report the mean accuracy along with the 95% confidence interval.

**Main Results**  Table 2 and Table 3 present the average accuracy rates for fine-grained learning from coarse labels on BREEDS and CIFAR-100. For each dataset, we report both 5-way and all-way fine-grained recognition results during testing. As shown in these tables, our proposed model is significantly better than other baseline methods on the above datasets. Especially on CIFAR-100, we achieve 4.05% and 6.36% improvements over state-of-the-art methods, even surpassing the fine upper-bound baseline significantly. Table 4 evaluates an intra-class case when all fine-grained classes of random coarse-grained classes were sampled in each episode. The results demonstrate that our method can better distinguish fine-grained categories in the embedding space.

**Retrieval Task Results**  In addition to evaluating the ability of learning fine-grained embeddings from coarse labels in few-shot learning tasks, Grafit [29] and MaskCon [8] also investigated the effectiveness of the 'coarse to fine' approach by using retrieval tasks for validation. As shown in Table 5, we conducted fair comparisons of our method with these state-of-the-art approaches, using

Table 5: Comparisons on CIFAR-100. Red and blue bold numbers are the top two best results.

| Methods | 5-way | All-way | Recall@1 | Recall@5 | mAP |
|---|---|---|---|---|---|
| Grafit (Reported in paper) | - | - | 60.57 | 82.32 | - |
| MaskCon (Reported in paper) | - | - | **65.52** | **83.64** | - |
| MaskCon (Our re-implementation) | **81.05** | **40.81** | 65.35 | 83.57 | **60.27** |
| Ours | **85.45** | **47.76** | **62.11** | **80.68** | **62.04** |

the same backbone and epoch settings as MaskCon. Our method still achieves superior results over Grafit and MaskCon (almost 7% improvement on all-way FSFG) under FSFG, and achieves a mean Average Precision (mAP) of 62.04%, surpassing MaskCon's 60.27%. This also reinforces the efficacy of our approach. Additionally, it is worth noting that our method is lower than MaskCon when we used Recall@k as an evaluation metric (by following MaskCon). The reason might be that image retrieval involves identifying relevant images from a gallery of diverse images, usually by utilizing a query image as a reference. In this scenario, the task entails finding any images belonging to the same category as the query image from the gallery, which might contain multiple images per category. On the other hand, few-shot recognition, exemplified by the one-shot scenario, presents a more challenging scenario. In this setting, the goal is to recognize objects/classes with very limited training examples, often relying on only a single image per category. This constraint inherently magnifies the difficulty of the task.

**Proposals in Our Method**  As shown in Table 6, we explored the improvement of each module in our method on LIVING-17 and CIFAR-100. The results of the first two rows show that the hyperbolic space is more suitable for the fine-grained embedding from coarse labels with a hierarchical structure, which is consistent with the conclusion shown in [11, 15]. The results of the last three rows show that the introduction of our HCM manner and AHCD strategy in the hyperbolic space can bring great improvement. This indicates that all proposals of our method can better learn the discriminative patterns between finer-grained categories from coarse labels.

Table 6: Comparisons of our proposals on LIVING-17 [23] and CIFAR-100 [16]. $\mathbb{E}$: Coarse-grained supervised learning in the Euclidean space. $\mathbb{H}$: Coarse-grained supervised learning in the hyperbolic space. HCM: Our proposed Hierarchical Cosine Margin manner. AHCD: Our proposed Adaptive Hierarchical Cosine Distance strategy.

| Methods | | | | LIVING-17 | | CIFAR-100 | |
|---|---|---|---|---|---|---|---|
| $\mathbb{E}$ | $\mathbb{H}$ | HCM | AHCD | 5-way | all-way | 5-way | all-way |
| ✓ | | | | 87.72 | 32.71 | 76.34 | 24.92 |
| | ✓ | | | 88.06 | 33.78 | 76.62 | 26.19 |
| | ✓ | ✓ | | 90.02 | 46.53 | 77.64 | 33.22 |
| | ✓ | ✓ | ✓ | 90.94 | 53.09 | 81.42 | 36.28 |

**Visualization**  As shown in Fig. 4, it can be observed that compared to training the model using only coarse labels, our method leads to a more concentrated distribution of sample points for each category, while also accentuating the distinctions between different classes. The visualization validates the effectiveness of our method of contributing to the discriminative power from the qualitative aspect.

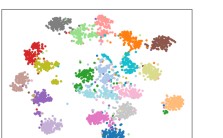 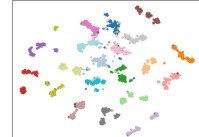

Figure 4: Visualization by $t$-SNE [13] on CIFAR-100. Left: training the model using only coarse labels. Right: our method.

**Adaptive Ability**  We extensively explored the effectiveness of the AHCD strategy. As shown in Fig. 5, we tracked the update processes of $d_1$ and $d_2$ during training on CIFAR-100 and LIVING-17 datasets. In general, there are the same trends and different details. In the early stages of training, as the model lacks discriminative ability, the values of $d_1$ and $d_2$ gradually decrease. As the model continues to train, it gradually acquires discriminative ability, resulting in an increase in the values of $d_1$ and $d_2$ followed by stabilization. From the final stable results, we observe that the margin between coarse-grained categories ($d_2$) is larger than the margin between fine-grained categories ($d_1$), and the margin between fine-grained categories on CIFAR-100 is larger than the margin between fine-grained categories on LIVING-17, which aligns with the true category relationships in the data distribution. This demonstrates that our proposed adaptive strategy can adjust $d_1$ and $d_2$ during training according to the actual data distribution.

**Hyperparameters**  As shown in Fig. 6, we changed the trade-off hyperparameter $\alpha$ of the hierarchical cosine loss in Eq. (8) on the CIFAR-100 and LIVING-17 datasets to observe its impact for the final result. It can be seen that, as $\alpha$ increases exponentially, the performance of the model increases

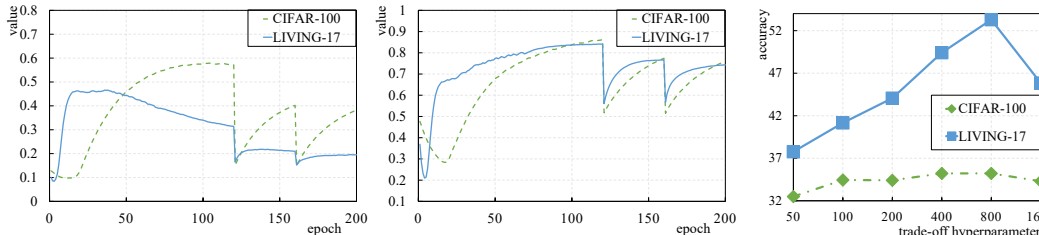

Figure 5: The update processes of $d_1$ (left) and $d_2$ (right) in adaptive hierarchical cosine distance strategy on CIFAR-100 and LIVING-17. $d_1$ and $d_2$ are the target distance for the same fine-grained and coarse-grained sample pairs.

Figure 6: Comparisons with different value of $\alpha$ on CIFAR-100 and LIVING-17.

first and then decreases. The value of $\alpha$ is also sensitive, with a smaller value of $\alpha$ limiting the discriminative hierarchical embedding ability of the HCM manner, and a larger value of $\alpha$ suppressing the generalization ability to fine-grained categories.

As shown in Fig. 7, we change the number of $k$-means clusters for each coarse-grained class in the HCM manner on CIFAR-100 and LIVING-17, and report the recognition results for all fine-grained categories (all-class) and intra-coarse class fine-grained categories (intra-class). From the experimental results, it can be observed that the number of clusters has a significant impact on the fine-grained recognition performance. When the number of clusters is small, the model performs poorly in fine-grained recognition tasks. Having a larger number of clusters than the actual number of fine-grained categories leads to better results. Additionally, in our experiments across five datasets, we found a recurring pattern that offers a guideline. Typically, when clustering coarse-grained categories, the number of clusters that yields optimal results is roughly twice the

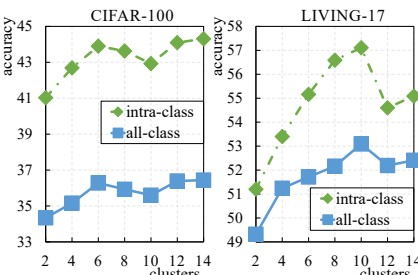

Figure 7: Comparison results in 1-shot learning with different numbers of clusters on CIFAR-100 and LIVING-17.

actual number of fine-grained subclasses within that coarse category. While this heuristic emerged consistently across our tested datasets, we acknowledge that it may not be universally optimal. However, it does provide a good starting point for users. In practice, fine-tuning based on validation performance is always recommended to ascertain the best hyperparameter for a specific task.

# 5    Conclusions

We proposed a novel method (PE-HCM) for fine-grained learning from coarse labels. We used the hyperbolic space to embed the samples and enhanced the discriminative ability with a hierarchical cosine margins manner. On the one hand, we performed supervised learning using coarse-grained labels to distinguish coarse-grained category samples initially. On the other hand, we constructed instance-level labels and fine-grained pseudo-labels using data enhancement and clustering methods, for the hierarchical cosine margins manner constraining the distance between sample pairs of four relationships, which resulted in a more refined hierarchical division of distances for fine-grained learning. By optimizing the hierarchical cosine distance using the hierarchical cosine margins manner, the learned embedding could be well generalized to fine-grained visual recognition tasks. Experiments on five benchmark datasets demonstrated the effectiveness of the proposed method.

Throughout our experiments, we have occasionally observed a particular scenario where our method yields comparatively lower results. This typically occurs when classes within the same hierarchy level exhibit substantial variations in their scopes. For instance, when some classes possess a significantly broader scope while others have a notably narrower scope. We believe this phenomenon arises from the fact that our method employs a uniform margin across all levels. In the future, more refined category-specific semantic hierarchical distance constraints are worth further study, that is, further constraining the distances between sample pairs of different categories within the same grain of granularity to make them more consistent with the real data distribution.

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
