# OpenReview forum: "Hyperbolic Space with Hierarchical Margin Boosts Fine-Grained Learning from Coarse Labels"
_NeurIPS.cc/2023/Conference — NeurIPS 2023 poster_

### Official Review · Reviewer_tnZ4 · 2023-07-06

**Soundness:** 3 good
**Presentation:** 3 good
**Contribution:** 2 fair
**Rating:** 4
**Confidence:** 4

**Summary:**

The paper adresses the issue of coarse to fine learning, _i.e._ coarse labels are present during training but the metric are computed on fine-grained labels. The paper introduces a method Poincaré embeddings with hierarchical cosine margins (PE-HCM). The paper suggests embedding the image representation in the hyperbolic space for coarse label classification to take advantage of its natural hierarchical organisation. It has a second branch that clusters with k-means instances that share the same coarse labels to simulate fine-grained pseudo-labels. It uses a hierarchical loss that aims at making the cosine distance between pairs match their relation, i.e. same instance label (same instance with different data augmentation), same cluster label (assigned by k-means), same coarse label (supervision), different coarse label (supervision). The paper validates experimentally the method on several benchmarks, CIFAR-100 and BREEDS - a collection of 4 datasets. The paper conducts ablation studies of the different components of the method.

**Strengths:**

S1. The method is well justified. The adaptive hierarchical distance is interesting and works well experimentally.

S2. Experimental results are encouraging.

S3. Ablation studies are present.

**Weaknesses:**

W1. Some works are missing in the related works: MaskCon (MaskCon: Masked Contrastive Learning for Coarse-Labelled Dataset, CVPR 2023) and Grafit (Grafit: Learning fine-grained image representations with coarse labels, ICCV 2021) that both tackle the coarse to fine task.  And UnHyp (Unsupervised Hyperbolic Metric Learning, CVPR 2021) that uses hyperbolic space to create a hierarchy.

W2. The paper lacks comparison to state-of-the-art methods such as MaskCon and Grafit. It could conduct experiments on the large scale iNaturalist-18 dataset.

W3. The paper lacks comparison to unsupervised baselines, such as UnHyp or more recent state-of-the-art self-supervised-learning methods, such as Dino (Emerging Properties in Self-Supervised Vision Transformers, ICCV 2021), iBot (iBOT: Image BERT Pre-Training with Online Tokenizer, ICLR 2022) etc.

Minor:

W4. The paper could remind the reader, in the benchmark datasets section, what classes are present in the different datasets, and what are the coarse classes on the different datasets.

**Questions:**

Q1. Line 145: The paper gives notations for coarse and fine-grained labels. Could the method take advantage of a hierarchy of classes rather than solely a coarse label?

Q2. What is the impact of the granularity of the coarse labels, _i.e._ how much will the performance deteriorate when the coarse labels become more and more broad and on the contrary what happens when given the fine-grained labels?

Q3. Line 227: the paper defines a distance that is used in the HCM module. It uses the Euclidian cosine distance, this is surprising as the paper states that it compares instances in the hyperbolic space. The paper should elaborate this design choice.

Q4. Table 2: reports results on the BREEDS benchmark. I am not sure to understand what are the results indicated as “5-way” and “all-way”.

Typos:

- Table 1: Coarase —> Coarse
- Table 5: European —> Euclidian

---

> ### Author Rebuttal · Authors · 2023-08-10
>
> ## Thank you for the positive comments. Below please find our responses to some specific comments.
>
> ### Comment_1:
>
> >  *Additional related works*
>
> Thanks for your kind reminder. These works do share some connections with our method. We did not notice them because these works have different task-of-interest from ours and are seldom discussed in prior few-shot fine-grained literature. We will cite the recommend works and add the discussions (as detailed in the following responses).
>
>
> ### Comment_2:
>
> >  *Comparison to SOTA methods such as MaskCon and Grafit*
>
> Thank you for the suggestion. Since MaskCon and Grafit are proposed for a different task (i.e., retrieval), we conduct a comparison on both our few-shot fine-grained (FSFG) task and the retrieval task with CIFAR-100, as shown in the table below.
>
>
> | Methods                     | 5-way (FSFG) | All-way (FSFG)|Recall@1 (Retrieval) | Recall@5 (Retrieval) |
> |-|-|-|-|-|
> | Grafit (Reported in paper)  | -     |-   |60.57    | 82.32    |
> | MaskCon (Reported in paper) | -     |-   |**65.52**    | **83.64**    |
> | MaskCon (Our Replication)   | 81.05     |40.81   | 65.35    | 83.57    |
> | Ours                         | **85.45**     |**47.76**   |62.11    | 80.68    |
>
> **Analysis:** As shown, under FSFG, our method still achieves superior results over MaskCon (almost 7% improvement on all-way FSFG). Under the retrieval scenario (their specialized task), our method already surpasses Grafit but is lower than MaskCon. The reason might be: MaskCon's self-supervised design emphasizes the finest granularity and thus is favored by instance-based retrieval. In contrast, our approach harmonizes embeddings across multiple granularities and thus is favored by FSFG.
>
> Given the time constraints, we couldn't finish the experiment on the large-scale iNaturalist-18 dataset in 7 days, but we pledge to incorporate results on this dataset in the final version.
>
> ### Comment_3:
>
> >  *Comparison to unsupervised baselines and more recent SOTA SSL methods*
>
> Thank you. We compare the two most recent unsupervised methods (DINO and iBot with their publicly released model) in the table below. It is observed that our method consistently outperforms both DINO and iBot across different metrics and datasets by a large margin.
>
> | Methods| CIFAR-100 || Living17|| Non-Living26||
> |-|-|-|-|-|-|-|
> || 5-way| All-way| 5-way| All-way| 5-way| All-way|
> | Dino-ResNet50 | 51.23 | 12.67 | 83.62  | 44.31 | 72.58 | 33.34 |
> | iBOT-ViT-Base16  | 46.72   | 10.68 | 75.41   | 31.38  | 74.89 | 32.95 |
> | Ours-ResNet50    | **81.42**    | **36.28** | **90.94**   | **53.09** | **89.97** | **50.12** |
>
> ### Comment_4:
>
> >  *The detailed classes in different datasets*
>
> Thanks. We adopt the protocol dataset split in their original papers. Briefly:
> - Living-17, rooted at "living thing" in WordNet with coarse-grained classes at a depth of 5, includes salamander, turtle, dog, bear, and monkey.
> - NonLiving-26, rooted at "non-living thing" and also at a depth of 5, encompasses bag, boat, car, digital computer, and ship.
> - Entity-13, originating from "entity" at a depth of 3, features bird, reptile, and mammal.
> - Entity-30, also rooted in "entity" but at depth 4, includes classes like serpentes, building, footwear, and fruit.
>
> Since the number of classes is large, we do not have enough space to enumerate all the class names. Please kindly refer to the original paper for details.
>
>
> ### Comment_5:
>
> > *Could the method take advantage of the hierarchy rather than solely a coarse label?*
>
>
> Our method already utilizes an estimated hierarchy. Specifically, given the coarse labels, we estimate the finer-grained classes through unsupervised clustering (Ln. 49) and form an estimated hierarchy (with inevitable noises). If we have access to the ground truth of fine-grained labels, the results are expected to be better.
>
> ### Comment_6
>
> > *Impact of the granularity of the coarse labels*
>
> Thanks for this good question. Generally, if the given labels become more fine-grained, the results become better. During rebuttal, we adjust the granularity by splitting the dataset into different classes (the larger number has finer granularity).
> The results are summarized below.
>
> |  | Living17 | | | NonLiving26 |||
> |-|-|-|-|-|-|-|
> | Num of Splitted Classes | 9        | 17    | 34    | 13          | 26    | 52    |
> | 5-way                         | 82.84    | 90.94 | 90.58 | 82.53       | 89.97 | 88.91 |
> | All-way                       | 43.65    | 53.09 | 63.94 | 42.51       | 50.12 | 54.78 |
> | Intra-class                   | 50.74    | 57.11 | 68.46 | 53.06       | 58.63 | 62.68 |
>
> Thank you for pointing out this new aspect. We believe this investigation provides valuable insight into our method, as well as the FSFG task itself.
>
> ### Comment_7
>
> > *Using Euclidean cosine distance in the hyperbolic space*
>
> The property of Conformality, as discussed in Ln. 181, plays a pivotal role by preserving the angles between feature vectors of instances within the hyperbolic space. This fundamental characteristic ensures that regardless of the choice of angular measurement, two feature vectors with the same angle will maintain that angle in the hyperbolic space. Consequently, employing the Euclidean cosine distance for this purpose is a valid approach. More importantly, during our experimental process, we observed that optimizing with the Euclidean cosine distance provided more stability in gradient computations compared to the hyperbolic cosine distance. This observation in the empirical results guided our choice to use the Euclidean cosine distance, as it aligned well with our objectives and offered more robust training. In the final version, we will follow your suggestion to include a more detailed explanation.
>
> ### Comment_8
>
> > *The meaning of "5-way" and "all-way"*
>
> "5-way" refers to tasks that involve 5 randomly selected fine-grained categories. "All-way", encompasses all available fine-grained categories in the dataset.

---

> > ### Author Response · Authors · 2023-08-15
> >
> > Dear Reviewer tnZ4,
> >
> > We would like to appreciate you again for your valuable comments. Moreover, it is important for us to know whether our responses have addressed your concerns, and we look forward to receiving your further feedback. In any case, we remain available to answer your questions.
> >
> > Best Regards,
> >
> > The authors

---

> > > ### Comment · Reviewer_tnZ4 · 2023-08-15
> > >
> > > I thank the authors for the detailed answers to my questions and taking the time to run experiments and replicate a new baseline, *MaskCon*. I have some final questions and comments on the authors' responses:
> > >
> > > A - Could you elaborate on the difference between FSFG and retrieval as done in *e.g.* MaskCon? Meaning why would a method be better on one task and not the other.
> > >
> > > B - Could you discuss the computational cost of using the Poincaré embeddings rather than the Euclidian ones?
> > >
> > > C - In addition to *Comment_5*, have you tried using more than one hierarchical level? For instance (when you will have time to run it, as said in *Comment_2*) this would be a nice experiment to have on iNat-18.
> > >
> > > D - Also what are the results if given access to the fine-grained labels? Does your method further improve the results or is it no more relevant in this particular case? (note that I understand that this not the setting that is claimed to be working)
> > >
> > > E - Have you measured some hierarchical metrics? Have you compared those with other methods, *e.g.* the "fine upper-bound"? This would quantitatively show that the good behaviour of your method comes from the good organisation of the embedding space.
> > >
> > > F - In relation to your response in *Comment_7*, how hard is it to train a network using hyperbolic space? You said that Euclidan space offers more stability when computing gradient, will practitioners need to tune a lot of hyper-parameters to make your method work?
> > > Is the choice of **c** (curvature) tricky? I do not believe you had an ablation for this hyper-parameter.

---

> > > > ### Author Response · Authors · 2023-08-21
> > > > **Response Part-1**
> > > >
> > > > Thank you for the comments and discussions. Below please find our responses to some specific comments.
> > > > ***
> > > > ***A** - Could you elaborate on the difference between FSFG and retrieval as done in e.g. MaskCon? Meaning why would a method be better on one task and not the other.*
> > > >
> > > > **Response-A**: Image retrieval involves identifying relevant images from a gallery of diverse images, usually by utilizing a query image as a reference. In this scenario, the task entails finding *any* image belonging to the same category as the query image from the gallery, which might contain multiple images per category. On the other hand, few-shot recognition, exemplified by the one-shot scenario, presents a more challenging scenario. In this setting, the goal is to recognize objects/classes with very limited training examples, often relying on only a single image per category. This constraint inherently magnifies the difficulty of the task.
> > > >
> > > > Our method excels in the FSFG task, where the inherent challenge lies in accurately recognizing objects/classes with minimal training instances. We have outperformed MaskCon in this domain, showcasing the effectiveness of our approach. Additionally, it is worth noting that in our previous reporting on retrieval, we primarily used Recall@k as an evaluation metric (by following MaskCon), which focuses on recall performance. However, when considering comprehensive metrics like mean Average Precision (mAP), our method achieves a mAP of 62.04%, surpassing MaskCon's 60.27%. This also reinforces the efficacy of our approach.
> > > > ***
> > > > ***B** - Could you discuss the computational cost of using the Poincaré embeddings rather than the Euclidian ones?*
> > > >
> > > > **Response-B**: The computational cost associated with utilizing Poincaré embeddings versus Euclidean embeddings is relatively comparable. In our method, the inclusion of clustering incurs a computational cost of $\mathcal{O}(n^2)$ within the Euclidean space. Similarly, in the context of Poincaré space, the computational cost also remains in the order of $\mathcal{O}(n^2)$, despite the additional mapping operations inherent to the Poincaré space representation.
> > > > ***
> > > > ***C** - In addition to Comment_5, have you tried using more than one hierarchical level? For instance (when you will have time to run it, as said in Comment_2) this would be a nice experiment to have on iNat-18.*
> > > >
> > > > **Response-C**: We introduced a coarser level of label hierarchy to the CIFAR100 and Living17 datasets. When conducting 1-shot 5-way and 1-shot all-way testing experiments, we observed a decrease in accuracy for the CIFAR100 dataset by 4.64% and 4.63% respectively. Similarly, the Living17 dataset exhibited reductions of 5.35% and 7.34%. This outcome underscores that our approach accounts for broader categories at the expense of finer-grained capabilities. Importantly, it validates that our method indeed learns from the provided labels rather than relying on a predefined hierarchy.
> > > > ***
> > > > ***D** - Also what are the results if given access to the fine-grained labels? Does your method further improve the results or is it no more relevant in this particular case? (note that I understand that this not the setting that is claimed to be working)*
> > > >
> > > > **Response-D**: Given access to the fine-grained labels, it further improves the results, i.e., +0.13% (5-way) and +3.78% (all-way) on CIFAR100 and +2.12% (5-way) and +4.63% (all-way) on Living17.
> > > > ***
> > > > ***E** - Have you measured some hierarchical metrics? Have you compared those with other methods, e.g. the "fine upper-bound"? This would quantitatively show that the good behaviour of your method comes from the good organisation of the embedding space.*
> > > >
> > > > **Response-E**: We would like to emphasize that our main contribution lies in leveraging the hierarchical structure within the Poincaré space to effectively exploit coarse labels to learn fine-grained features. We have demonstrated the advantages of introducing the margin concept commonly utilized in Euclidean spaces into the Poincaré space, and its beneficial impact on our method's performance. While we have incorporated an estimated hierarchy, it is important to note that the utilization of hierarchy is not the sole focus of our method. Our method's core innovation lies in its adeptness at harnessing the Poincaré embedding space for enhanced recognition tasks. It is worth considering that other methods that specialize in utilizing hierarchical structures might indeed be compatible with and complementary to our approach. Comparison and cooperation with other good methods in Euclidean space deserve future investigation.

---

> > > > > ### Author Response · Authors · 2023-08-21
> > > > > **Response Part-2**
> > > > >
> > > > > ***F** - In relation to your response in Comment_7, how hard is it to train a network using hyperbolic space? You said that Euclidan space offers more stability when computing gradient, will practitioners need to tune a lot of hyper-parameters to make your method work? Is the choice of c (curvature) tricky? I do not believe you had an ablation for this hyper-parameter.*
> > > > >
> > > > > **Response-F**: For practical usage, we follow previous work to adopt the popular used setting for the hyperparameters in experiments. However, when employing the hyperbolic space for training, we encountered instances of NaN losses during the training process. In contrast, training within the Euclidean space yielded stable outcomes for our network. As a result, practitioners will find it unnecessary to extensively fine-tune hyperparameters to achieve successful method execution. Regarding the choice of curvature, it is also non-tricky, as we have adopted the default configuration in accordance with [Ref1]. Notably, in Fig. 5 and Fig. 6 of the paper, we had indeed conducted ablation studies for the hyper-parameters in our method.
> > > > >
> > > > > [Ref1] Hyperbolic image embeddings, CVPR 2020.

---

### Official Review · Reviewer_EUtT · 2023-07-06

**Soundness:** 3 good
**Presentation:** 4 excellent
**Contribution:** 4 excellent
**Rating:** 8
**Confidence:** 5

**Summary:**

This paper proposes a method developed in the hyperbolic space to embed visual embeddings to deal with the task of fine-grained learning from coarse labels. By discovering the advantages of hyperbolic space, the authors design the hierarchical cosine margin manner and an adaptive hierarchical cosine distance, which favour modelling fine-grained objects. Experiments are conducted on five benchmark datasets. Evaluations show that the proposed method achieves superior recognition accuracy over competing solutions on these datasets.

**Strengths:**

+ The proposed method is novel and makes sense, which has great technical contributions, especially for the introduction of hyperbolic space into the fine-grained learning from coarse labels.

+ The fine-grained learning from coarse labels task has good practical potential, which also is a challenging problem in computer vision and machine learning. Such a task deserves further exploration.

+ The hierarchical cosine margins and the adaptive hierarchical cosine distance are interesting. These modules are crucial for the whole framework of the proposed method and they are also tailored designs for the task studied in this paper.

+ The experiments are comprehensive and convincing. Overall, they can validate the effectiveness of the proposed method, as well as the proposed modules.

+ The paper is well-written and easy to follow.


**Weaknesses:**

Although the motivation of Figure 1 is relatively clear, it can be further explained.

**Questions:**

- As stated in the section of Hyperparameters, different numbers of clusters lead to different accuracy. How to choose an appropriate hyperparameter for a specific task for users?

- Some minor issues of paper writing should be fixed, e.g., “Fig” should be fixed as “Fig.” or “Figure”.


**Limitations:**

This paper does not explicitly discuss its limitations.

---

> ### Author Rebuttal · Authors · 2023-08-10
>
> ## Thank you for the positive comments. Below please find our point-to-point responses.
>
> ### Comment_1:
>
> > *Explain the motivation of Figure 1 in more details*
>
> Thanks for your suggestion. Figure 1 primarily illustrates the properties of the Poincaré disk space and the impact of cosine distance constraints on the mapped space. One intrinsic characteristic of this mapping, critical for our work, is the Conformality property derived from Riemannian geometry. It ensures that angles between curves or vectors in the Euclidean space remain consistent when projected onto the Poincaré disk.
>
> By emphasizing this property in Figure 1, we intend to provide a clearer comprehension of the principles underpinning our method. Detailing this geometric phenomenon underscores the robust theoretical base of our approach and lends deeper insight into our motivation. In our revised manuscript, we will ensure this aspect is delineated more prominently.
>
> ### Comment_2:
>
> > *Selection of appropriate hyperparameter*
>
> Thank you for pointing out the importance of choosing appropriate hyperparameters. In our experiments across five datasets, we found a recurring pattern that offers a guideline. Typically, when clustering coarse-grained categories, the number of clusters that yields optimal results is roughly twice the actual number of fine-grained subclasses within that coarse category.
>
> While this heuristic emerged consistently across our tested datasets, we acknowledge that it may not be universally optimal. However, it does provide a good starting point for users. In practice, fine-tuning based on validation performance is always recommended to ascertain the best hyperparameter for a specific task. We will clarify this heuristic in our revised manuscript to guide potential users of our method.
>
> ### Comment_3:
>
> > *Minor issues in paper writing*
>
> Thank you for highlighting this inconsistency. We will align the use of "Fig." and "Figure"  throughout the paper in the revised version. We appreciate your attention to detail and will carefully proofread the manuscript.

---

> > ### Comment · Reviewer_EUtT · 2023-08-13
> > **After rebuttal**
> >
> > After reviewing the author's response and considering the feedback from other reviewers, I have decided to maintain my initial score of "strong accept". I believe this is a solid paper.

---

### Official Review · Reviewer_FRH5 · 2023-07-09

**Soundness:** 3 good
**Presentation:** 3 good
**Contribution:** 3 good
**Rating:** 7
**Confidence:** 4

**Summary:**

The paper presents a novel method, PE-HCM, for fine-grained learning from coarse labels. The authors propose the use of the hyperbolic space for sample embedding and introduce a hierarchical cosine margins manner to enhance discriminative ability. The method combines supervised learning with coarse-grained labels and instance-level labels obtained through data enhancement and clustering. Experimental results on benchmark datasets demonstrate the effectiveness of the proposed approach.

**Strengths:**

* The paper addresses an important problem in the field of fine-grained learning, which is the challenge of learning from coarse labels. Fine-grained recognition is crucial in various applications, and the proposed method offers a solution to improve the performance in this context. The significance of the problem adds value to the research presented in the paper.
* The paper introduces several novel concepts and methods. The use of the hyperbolic space for sample embedding is a unique approach that has shown promise in other related works. Additionally, the introduction of the hierarchical cosine margins manner to enhance discriminative ability is an innovative strategy that contributes to refining the hierarchical divisions for fine-grained recognition. The combination of these concepts and methods in the proposed PE-HCM method demonstrates the originality of the research.
* The paper exhibits good writing quality, characterized by clarity and soundness. The paper effectively communicates the motivation, methodology, and experimental results to the readers. The descriptions of the proposed method and the experimental setup are clear and well-presented. The writing style contributes to the overall understanding of the research and enhances the readability of the paper.
* The paper demonstrates a high level of experimental rigor. The authors provide detailed descriptions of the benchmark datasets used, including their characteristics and sizes. They also compare the proposed method with several state-of-the-art models, covering a wide range of relevant approaches in the field of embedding learning. By conducting extensive experiments on multiple benchmark datasets, the authors validate the effectiveness of their proposed method and show its superiority over existing techniques. The inclusion of statistical measures such as mean accuracy and confidence intervals further enhances the reliability of the experimental results.
* The paper showcases the adaptive ability of the proposed method through the introduction of the Adaptive Hierarchical Cosine Distance (AHCD) strategy. The authors track the update processes of the distance parameters and show that the method adjusts these parameters during training to align with the actual data distribution. Furthermore, the paper provides a thorough analysis of the trade-off hyperparameter α, demonstrating the sensitivity of the method to its value and offering insights into finding an appropriate balance between discriminative hierarchical embedding and generalization ability.


**Weaknesses:**

* While the paper includes theoretical illustrations of sample pair distances, it would be beneficial for the authors to provide empirical evidence by visualizing the feature distances between real samples.
* The paper focuses on the evaluation of the proposed method on specific benchmark datasets but lacks a broader discussion on the generalizability of the results. Providing insights into the transferability of the proposed approach to other fine-grained recognition tasks or datasets would strengthen the overall impact of the paper.


**Questions:**

The paper introduces the use of the hyperbolic space for embedding samples and applies the hierarchical cosine margins to enhance the discriminative ability. While the results demonstrate improved performance in fine-grained recognition, I would like to inquire about the interpretability of the learned embeddings. Can the authors provide more insights into the interpretability of the hyperbolic embeddings and how the hierarchical cosine margins contribute to the separability of fine-grained categories? It would be helpful to include visualizations or case studies that illustrate how the learned embeddings capture meaningful hierarchical relationships and contribute to the discriminative power.

**Limitations:**

The paper concludes by discussing potential directions for future research, highlighting the importance of refining category-specific semantic hierarchical distance constraints. This suggestion indicates the authors' awareness of the limitations of the proposed method and opens up opportunities for further exploration and improvement in fine-grained learning from coarse labels.

---

> ### Author Rebuttal · Authors · 2023-08-10
>
> ## Thank you for the positive comments. Below please find our point-to-point responses.
>
> ### Comment_1:
>
> > *Visualization and interpretability*
>
> Thank you for your suggestion. We have provided the visualization by t-SNE on CIFAR-100 in the global rebuttal file. As shown, it can be observed that compared to training the model using only coarse labels, our method leads to a more concentrated distribution of sample points for each category, while also accentuating the distinctions between different classes. The visualization validates the effectiveness of our method of contributing to the discriminative power from the qualitative aspect. We will also add it in the final version.
>
> ### Comment_2:
>
> > *Broader discussion*
>
> Thanks for this suggestion. Our proposed approach, rooted in the principles of hyperbolic embeddings and hierarchical cosine margins, is intrinsically designed to capture hierarchical relationships. Such hierarchies are ubiquitous in nature and span various tasks beyond fine-grained recognition, encompassing a broad spectrum of datasets and domains.
>
> While this study focuses on specific benchmarks, we acknowledge the reviewer's point regarding the potential applicability of our method to a diverse set of tasks influenced by natural hierarchies. In our future work, we intend to explore the versatility of our method in tasks characterized by hierarchical structures. By expanding our experimentation horizon, we will delve into the broader potentials and possible limitations of our approach.
>
> We genuinely appreciate your feedback and believe that broadening the applicability of our method in subsequent research will shed light on its capabilities in handling tasks with inherent hierarchical relationships, thereby increasing its impact and significance in the broader scientific community.

---

> > ### Comment · Reviewer_FRH5 · 2023-08-19
> >
> > Thanks for the author's response. I would also maintain my initial score of accept.

---

### Official Review · Reviewer_zkc6 · 2023-07-12

**Soundness:** 3 good
**Presentation:** 3 good
**Contribution:** 3 good
**Rating:** 7
**Confidence:** 3

**Summary:**

This paper addresses the challenge of learning fine-grained embeddings from coarse labels, where detailed distinctions required for fine-grained tasks are often lacking. The authors propose a novel method that embeds visual embeddings into a hyperbolic space and enhances their discriminative ability using a hierarchical cosine margins approach. The hyperbolic space offers advantages such as capturing hierarchical relationships and increased expressive power, making it suitable for modeling fine-grained objects. By enforcing relatively large/small similarity margins between coarse/fine classes in the hyperbolic space, the proposed hierarchical cosine margins approach is introduced. While enforcing similarity margins in the regular Euclidean space is popular for deep embedding learning, extending it to the hyperbolic space is non-trivial and valuable for coarse-to-fine generalization. Experiments are conducted on five benchmark datasets.

**Strengths:**

1.	Novel Method: The paper introduces a novel method, Poincare embedding with hierarchical cosine margins (PE-HCM), which addresses the challenging task of fine-grained learning from coarse-grained labels. This innovative approach bridges the gap between coarse-grained and fine-grained labels, enabling the transfer of knowledge from coarse-grained categories to fine-grained recognition tasks.
2.	Technical Contribution: The proposed method incorporates key technical contributions. It utilizes the hyperbolic space to capture hierarchical relationships and provide increased expressive power. Additionally, the hierarchical cosine margins enforce proximity relationships among sample pairs, enabling the model to learn fine-grained features and distinctions. The adaptive strategy for updating target distances further enhances the model's ability to capture underlying relationships and adapt to the fine-grained characteristics of the data.
3.	Paper Writing: The paper is well-written and effectively communicates the motivations, challenges, and proposed solutions. The introduction provides a clear background and problem statement, leading to a comprehensive description of the proposed method. The technical details, including the use of hyperbolic space and hierarchical cosine margins, are explained concisely, making the paper accessible to readers.
4.	Experimental Performance: The proposed method is thoroughly evaluated on five benchmark datasets, demonstrating its superior performance compared to competing solutions. By achieving state-of-the-art recognition accuracy, the experimental results highlight the effectiveness and potential practical applications of the proposed method in fine-grained recognition tasks.


**Weaknesses:**

1.	Limited Analysis of Failure Cases: The paper could include a discussion of the failure cases of the proposed method. Analyzing scenarios where the method does not perform well would provide insights into potential weaknesses and opportunities for future improvements.

2.	Minor Issues: There are several typos throughout this paper, e.g., in Table 1, Coarase should be Coarse.



**Questions:**

I am curious about how does the proposed method perform when the available coarse-grained labels are noisy or contain inaccuracies?

---

> ### Author Rebuttal · Authors · 2023-08-10
>
> ## Thank you for the positive comments. Below please find our responses to some specific comments.
>
> ### Comment_1:
>
> >  *Limited analysis of failure cases*
>
> Thanks for this suggestion. We fully concur that a deep dive into the scenarios where our model underperforms can provide valuable insights and pave the way for future refinements.
>
> During our experiments, we used to observe a typical case: if the classes in the same level have very different scopes (e.g., some classes have very large scope and some classes have very small scope), our method tends to achieve relatively low results. We infer it is because our method uses the same margin for each level. We will add the discussion into the manuscript. We also believe that such investigation will help our future research.
>
>
> ### Comment_2:
>
> > *Several typos*
>
> Thank you for pointing out the typographical errors. We will meticulously review the manuscript to correct these typos, including the one you've highlighted in Table 1, and ensure the clarity and professionalism of the revised submission.
>
> ### Comment_3:
>
> > *How does the proposed method perform when the available coarse-grained labels are noisy or contain inaccuracies*
>
> Thank you for this good question. Adding noises to the coarse-grained labels considerably compromises the accuracy. During rebuttal, we randomly add 20% noises to the ground-truth labels on CIFAR100 dataset, and observe a decrease of 10.28% 5-way accuracy. Investigating the noise problem is of realistic value and may point the new direction for our future work.

---

> > ### Comment · Reviewer_zkc6 · 2023-08-19
> >
> > Thanks for the author's response.  I would maintain my initial score of accept.

---

### Author Rebuttal · Authors · 2023-08-10

We thank all the reviewers for their valuable comments. We provide point-to-point responses to each reviewer, as well as a supplmentary PDF for some visualization results.

---

### Decision · Program_Chairs · 2023-09-21

**Decision:**

Accept (poster)

**Comment:**

This paper proposes a method that leverages hyperbolic space to embed visual representations for addressing the task of fine-grained learning based on coarse labels. The effectiveness of the proposed approach is demonstrated through experiments on benchmark datasets. The majority of the reviewers acknowledge the novelty of this paper and the comprehensiveness of the experiments. Detailed responses and discussions were provided regarding some of the shortcomings pointed out by the reviewers, such as the absence of comparisons with certain baseline methods.
Overall, the proposal is technically reasonable and has a certain enlightening significance for the utilization of label relationships. The experiment was sufficient, including multiple fine-grained learning benchmark datasets and existing SOTA methods, and the proposed method achieved consistent improvements. Therefore, I am inclined to recommend accepting this paper.